# Poor outcome of pediatric patients with acute myeloid leukemia harboring high FLT3/ITD allelic ratios

Kun-yin Qiu[1,2,4], Xiong-yu Liao[1,2,4], Yong Liu[1,2,3,4], Ke Huang[1,2], Yang Li[1,2], Jian-pei Fang 🄳 [1,2✉] & Dun-hua Zhou 🄳 [1,2✉]

Activating FLT3 mutations are the most common mutations in acute myeloid leukemia (AML), but the optimal threshold of FLT3/ITD allelic ratio (AR) among pediatric AML patients remains controversial. Here, we present the outcome and prognostic significance of FLT3/ITD AR analysis among pediatric patients with AML from the TARGET dataset. Applying fitting curve models and threshold effect analysis using the restrictive cubic spline function following Cox proportional hazards models identifies the cut-off value of 0.5 on FLT3/ITD AR. Moreover, we observe that high FLT3/ITD AR patients have an inferior outcome when compared to low AR patients. Our study also demonstrates that stem cell transplantation may improve the outcome in pediatric AML patients with high FLT3/ITD AR and may be further improved when combined with additional therapies such as Gemtuzumab Ozogamicin. These findings underline the importance of individualized treatment of pediatric AML.

[1] Children's Medical Center, Sun Yat-sen Memorial Hospital, Sun Yat-sen University, Guangzhou 510120, PR China. [2] Guangdong Provincial Key Laboratory of Malignant Tumor Epigenetics and Gene Regulation, Sun Yat-sen Memorial Hospital, Sun Yat-sen University, Guangzhou 510120, PR China. [3] Hematologic Laboratory of Pediatrics, Sun Yat-sen Memorial Hospital, Sun Yat-sen University, Guangzhou 510120, PR China. [4]These authors contributed equally: Kun-yin Qiu, Xiong-yu Liao, Yong Liu. ✉email: fangjpei@mail.sysu.edu.cn; zhoudunh@mail.sysu.edu.cn

Activating mutations of the FMS-related tyrosine kinase 3 (FLT3) receptor gene leads to constitutive activation of the FLT3 receptor tyrosine kinase and results in autonomous, cytokine-independent proliferation in vitro, which are the most common somatic mutations observed in acute myeloid leukemia (AML)[1]. It has been reported that the incidence of FLT3/ITD in adult AML is about 20–35%, whereas pediatric AML has a reported prevalence of 15% for FLT3/ITD, which was lower than in adults[2–12]. Previous reports demonstrated that childhood patients who harbored mutant FLT3/ITD had poor AML prognosis in certain large-scale studies[10,13–15]. Moreover, a high allelic ratio (AR) among patients with mutant FLT3/ITD to wild-type alleles revealed poor prognosis among pediatric and adult AML patients[16–21]. It is known that different ARs of FLT3/ITD demonstrated uneven prognostic significance, and a crucial cut-off value of 0.5 for FLT3/ITD AR has been well established in European LeukemiaNet (ELN 2017)[22], which has been reclassified as a favorable risk, but is only restricted to adult AML, whereas the threshold for pediatric AML remains controversial.

In this present study, we report a comprehensive study among childhood AML to explore the characterization of FLT3-ITD AR and its influence on outcomes. Most importantly, our primary purpose was to determine an optimal cut-off value for FLT3-ITD AR in a large-scale study of 1857 pediatric AML patients from the Therapeutically Applicable Research to Generate Effective Treatments (TARGET) database in order to provide evidence for risk-stratified therapy and prognostic evaluation in the FLT3/ITD-positive pediatric patient population.

## Results

**Baseline characteristics of the study population.** In total, 1857 pediatric AML patients were enrolled in this study. Among the TARGET group, 974 (52.5%) were male and 883 (47.5%) were female, and the median age in our cohort was 9.5 years old. The demographic, laboratory, and clinical characteristics of pediatric AML patients were compared based on the FLT3/ITD status (Table 1). The prevalence of FLT3/ITD mutation in the whole cohort was 18.4%. Mutant FLT3/ITD was more common in males compared with wild-type in terms of gender distribution (57.8% vs. 51.3%, $P = 0.029$). The median age of mutant FLT3/ITD was higher than that of wild-type (11.9 years vs. 8.4 years, $P < 0.001$) and mutant FLT3/ITD was far more common in older patients (å 10 years) (66.6% of patients harbored mutant FLT3/ITD, compared with 43.5% of patients with wild-type). The initial median WBC of mutant FLT3/ITD patients was higher than that of wild-type patients ($70.6 \times 10^9$/L vs. $25.7 \times 10^9$/L, $P = 0.004$), and was more common in patients with WBC > $50 \times 10^9$/L (60.1% vs. 34.7%, $P = 0.011$). Both peripheral blood (PB) blasts and bone marrow (BM) blasts among children harboring FLT3/ITD mutation were significantly higher than those harboring wild type (PB blast: 65% vs. 41%, $P < 0.001$; BM blast: 80% vs. 68%, $P < 0.001$).

52.6% of childhood AML harbored mutant FLT3/ITD accompanied by normal karyotype, compared with 18.4% which harbored wild-type ($P < 0.001$). FLT3/ITD mutation patients were mainly observed in the high-risk group (65.8%) and only 3.2% pediatric AML harbored wild-type in the high-risk group. The incidence of coexistence of mutant FLT3/ITD and WT1 mutation was 21.8%, which was higher than those which harbored wild-type and WT1 mutation ($P < 0.001$), where similar results also appeared in the coexistence of mutant FLT3/ITD and NPM1 mutation (16.9% vs. 6.5%, $P < 0.001$). Patients with mutant FLT3/ITD were more likely to have the FAB M1 phenotype than those with wild-type (25% vs. 10%; $P < 0.001$). None of the patients who harbored FLT3/ITD mutations had FAB M7 (Table 1). Within

the cohort of the high-risk group, patients with mutant FLT3/ITD had significantly higher prevalence than patients with wild-type (65.8% vs. 3.2%; $P < 0.001$).

**Clinical outcome and prognostic impact of FLT3/ITD mutations.** At the end of course 1 of induction therapy, 215 (64.6%) of the 341 participants with FLT3/ITD mutations achieved a CR compared with 1154 (77.2%) of 1516 participants with wild-type ($P < 0.001$). After course 2 of induction therapy, childhood AML with mutant FLT3/ITD had an inferior induction treatment response where 245 (77.8%) participants achieved CR, as opposed to 1285 (88.8%) cases with wild-type (Table 1, $P < 0.001$).

The median follow-up time (and range) for childhood AML alive at last contact was 2.9 (0.1–10.9) years for patients with mutant FLT3/ITD and 3.6 (0.1–11.3) years for those with wild-type ($P < 0.001$). EFS(SD) at 5 years from study entry for pediatric AML patients which harbored mutant FLT3/ITD was 36.6% (11.6%) compared with 45.6% (5.5%) for the FLT3/ITD wild-type population ($P < 0.001$; Fig. 1A). Similarly, 5-year OS (SD) of childhood AML patients with mutant FLT3/ITD were inferior to patients with wild-type (50.1% (10.7%) vs 63.3% (4.8%), $P < 0.001$; Fig. 1B).

**The threshold of FLT3/ITD AR among pediatric AML.** In this current study, FLT3/ITD AR varied from 0.01 to 15.8, with a median of 0.55, with 44 participants having AR greater than 1.0. To determine the optimal threshold of AR, we initially assessed the FLT3/ITD AR as a continuous variable for adverse clinical outcome from research entry.

To explore the relationship between FLT3/ITD AR and the risk of all-cause mortality in AML patients, we performed a fitting curve analysis by using the restrictive cubic spline function following on Cox proportional hazards models. This analysis was performed using log-converted and unconverted data. Logarithms (relative risk) can be converted to relative risk by pairing logarithms. After adjusting for factors that may be associated with all-cause mortality, including gender, chemotherapy protocol; risk group; BM blast; PB blasts; CNSL; FAB category; karyotype; WBC group; age group; CEBPA status; WT1 status; NPM1 status, FLT3/ITD AR was found to have a biphasic distribution (Fig. 2), with an initial rapid increase in the low AR (the value of AR < 0.5), while that was followed by a gradual linear increase in the high AR greater than 0.5. Furthermore, a risk of all-cause mortality seemed to rise with increasing AR after the turning point (AR ≥ 0.5).

To more accurately identify the cut-off value of 0.5 on FLT3/ITD AR that can distinguish patients with FLT3/ITD at high risk from the low-risk patients, we performed threshold effect analysis using Cox proportional hazards models. The threshold effect of AR on all-cause mortality was significant after adjusting for potential confounders. The adjusted regression coefficient (LogRR) was 45.1 (95%CI: 0.9–2281.6, $P = 0.057$) for AR < 0.5 while 1.6 (95% CI: 1.1–2.3, $P = 0.013$) for AR ≥ 0.5 (Table 2). Therefore, FLT3/ITD AR of 0.5 was selected as the prognostic threshold for further comparison. FLT3/ITD High AR was defined as AR ≥ 0.5, and low AR was defined as AR < 0.5.

**Clinical implication of FLT3/ITD AR.** We compared the laboratory and clinical characteristics of patients with high and low FLT3/ITD AR (Table 3). Those with high AR had a significantly elevated WBC count with a median WBC count of $92.4 \times 10^9$/L, compared with $49.2 \times 10^9$/L for the patients with low AR ($P < 0.001$). Patients with high AR had a significantly higher BM and PB blast percentage of 85% and 73% compared with that of 69% and 49% for patients with low AR (both of $P < 0.001$),

**Table 1 Baseline characteristics of study participants by FLT3/ITD status classification.**

| Characteristics | Total | FLT3/ITD status | | P value |
|---|---|---|---|---|
| | | Wild type (n = 1516) | Mutation (n = 341) | |
| Gender, n (%) | | | | 0.029 |
| Male | 974 (52.5%) | 777 (51.3%) | 197 (57.8%) | |
| Female | 883 (47.5%) | 739 (48.7%) | 144 (42.2%) | |
| Age group (y) | | | | <0.001 |
| <10 | 970 (52.2%) | 856 (56.5%) | 114 (33.4%) | |
| ≥10 | 887 (47.8%) | 660 (43.5%) | 227 (66.6%) | |
| FAB category | | | | <0.001 |
| M0 | 25 (3.3%) | 23 (3.8%) | 2 (1.3%) | |
| M1 | 99 (13.1%) | 60 (10.0%) | 39 (25.0%) | |
| M2 | 205 (27.2%) | 163 (27.2%) | 42 (26.9%) | |
| M4 | 217 (28.7%) | 173 (28.9%) | 44 (28.2%) | |
| M5 | 163 (21.6%) | 137 (22.9%) | 26 (16.7%) | |
| M6 | 14 (1.9%) | 11 (1.8%) | 3 (1.9%) | |
| M7 | 32 (4.2%) | 32 (5.3%) | 0 (0.0%) | |
| Chemotherapy protocol, n (%) | | | | 0.001 |
| AAML03P1 | 111 (6.0%) | 91 (6.0%) | 20 (5.9%) | |
| AAML0531 | 725 (39.0%) | 570 (37.6%) | 155 (45.5%) | |
| AAML1031 | 954 (51.4%) | 808 (53.3%) | 146 (42.8%) | |
| CCG-2961 | 67 (3.6%) | 47 (3.1%) | 20 (5.9%) | |
| WBC group, n (%) | | | | <0.001 |
| <50 × 10$^9$/L | 1125 (60.6%) | 989 (65.3%) | 136 (39.9%) | |
| ≥50 × 10$^9$/L | 731 (39.4%) | 526 (34.7%) | 205 (60.1%) | |
| Risk group, n (%) | | | | <0.001 |
| Low risk | 665 (36.6%) | 621 (42.0%) | 44 (13.1%) | |
| Standard risk | 881 (48.5%) | 810 (54.8%) | 71 (21.1%) | |
| High risk | 269 (14.8%) | 48 (3.2%) | 221 (65.8%) | |
| CR status at end of course 1 | | | | <0.001 |
| CR | 1369 (74.9%) | 1154 (77.2%) | 215 (64.6%) | |
| Not in CR | 423 (23.2%) | 312 (20.9%) | 111 (33.3%) | |
| Death | 35 (1.9%) | 28 (1.9%) | 7 (2.1%) | |
| CR status at end of course 2 | | | | <0.001 |
| CR | 1530 (86.8%) | 1285 (88.8%) | 245 (77.8%) | |
| Not in CR | 190 (10.8%) | 128 (8.8%) | 62 (19.7%) | |
| Death | 42 (2.4%) | 34 (2.3%) | 8 (2.5%) | |
| SCT in 1st CR | | | | <0.001 |
| No | 1414 (82.7%) | 1233 (87.5%) | 181 (60.3%) | |
| Yes | 295 (17.3%) | 176 (12.5%) | 119 (39.7%) | |

The qualitative data were analyzed using the chi-square test and the quantitative data were compared using the Student's t-test (two-tailed).
CR complete remission, SCT stem cell transplantation.

respectively. In addition, cases with high AR were more common to have FAB M1 than those with low AR (35.7% vs. 12.5%; $P < 0.001$). Those with high AR had a CR rate of 72.5% vs. 84% for those with low AR at the end of course 2 ($P = 0.048$). As expected, the proportion of pediatric AML with high AR who accepted SCT was significantly higher than those with low AR (47.2% vs. 30.9%, $P = 0.004$). Meanwhile, patients harboring high FLT3/ITD AR were more likely to appear in the high-risk group with a percentage of 65.8%, which was higher than those harboring low AR with only a percentage of 3.2% ($P < 0.001$). However, GO treatment seems to be similar in the distribution between high and low AR group (55.4% vs. 49.4%, $P = 0.479$).

Subsequently, K–M survival analysis was performed to determine if an AR value of 0.5 was suitable to establish a prognostic threshold for classifying FLT3/ITD populations into those at high risk for treatment failure and those who were not. Corresponding EFS at 5 years for those with high AR and low AR was 34.1% (15.6%) and 39.7% (17.1%) ($P = 0.042$; Fig. 1C), respectively. Overall survival at 5 years for patients with high AR and low AR was 48.3 (14.6%) and 54.2 (15.5%), respectively ($P = 0.049$; Fig. 1D). This study confirmed our findings that the FLT3/ITD AR of 0.5 is a clinically useful risk identification threshold in the FLT3/ITD population.

**The impact of SCT in pediatric AML with FLT3/ITD mutation**. Of the 295 SCT recipients, 119 (40.3%) had mutant FLT3/ITD and the remaining 176 (59.7%) had wild-type. We initially compared outcomes directly between FLT3/ITD wild-type patients receiving SCT ($n = 176$) and chemotherapy-only ($n = 1233$). In FLT3/ITD wild-type patients, EFS at 5 years was 52.3% (21.4%) for the recipients of SCT versus 48.1% (5.9%) for those who received chemotherapy-only ($P = 0.11$; Fig. 3A). Corresponding OS at 5 years was 55.6% (14%) vs. 66.3% (4.9%) for SCT and chemotherapy recipients, respectively ($P = 0.015$; Fig. 3B). Subsequently, we further compared the outcomes of patients with FLT3/ITD mutation who received SCT ($n = 119$) with those treated with chemotherapy-only ($n = 181$). Within the cohort harboring FLT3/ITD mutations, SCT showed a favorable outcome on 5-year EFS (58.6% (18.1%) vs. 28.3% (15.1%); $P < 0.001$) and a better 5-year OS (64.3% (16.2%) vs. 45.7% (15%); $P < 0.001$) than those with chemotherapy-only (Fig. 3C, D).

Among the 119 cases with mutant FLT3/ITD who received SCT, 76 had high and 43 had low FLT3/ITD AR, whereas of the 181 patients with FLT3/ITD who received conventional chemotherapy, 85 had high AR and 96 had low AR. In patients with chemotherapy-only who harbored high FLT3/ITD AR, 5-year EFS was 14.5% (18.3%), compared with 39.8% (21.8%) in low

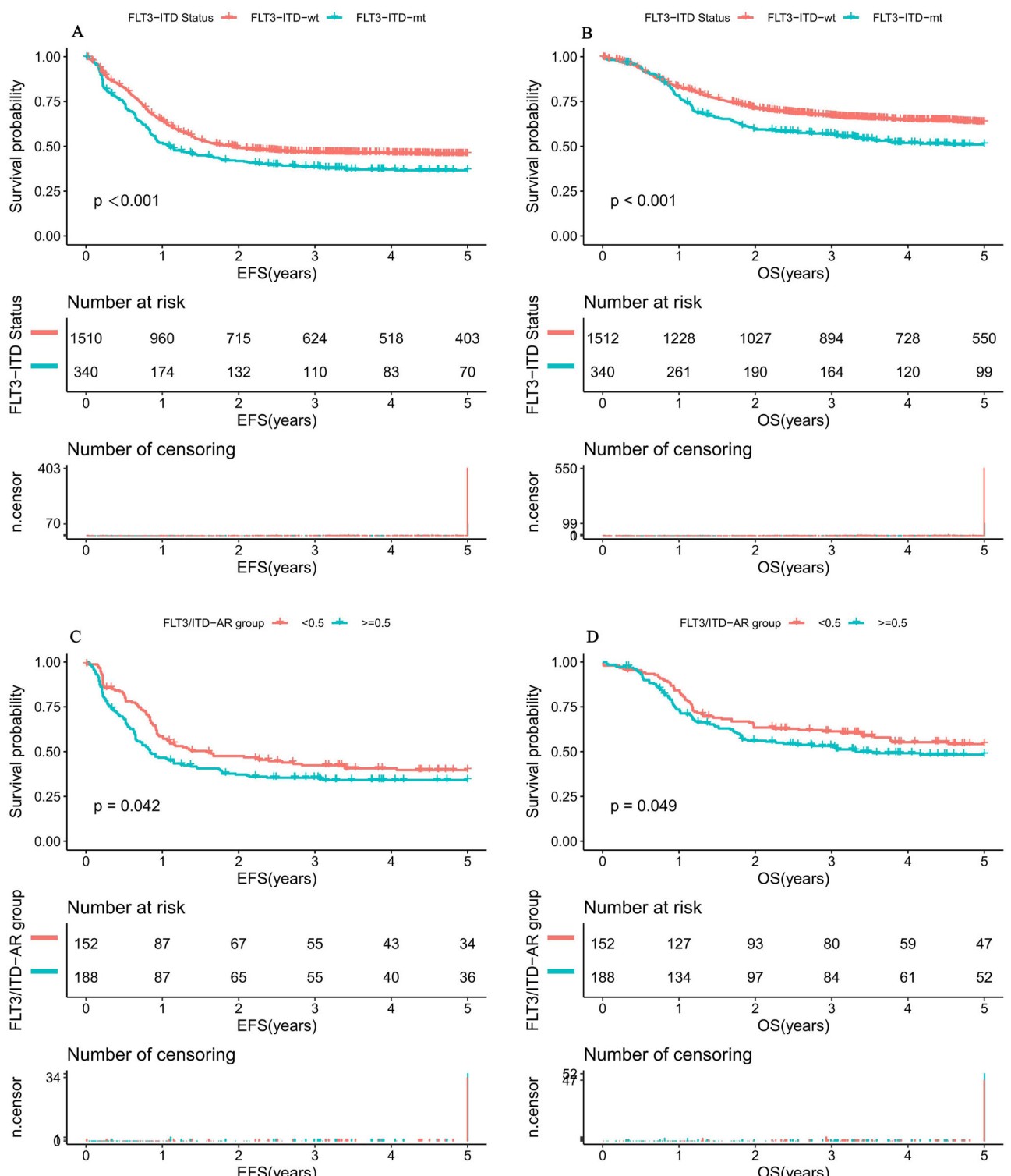

**Fig. 1 Survival curves of pediatric AML patients with and without FLT3/ITD mutations and with different AR. A** Probability of EFS for all patients with and without FLT3/ITD mutations. **B** Probability of OS for all patients with and without FLT3/ITD mutations. **C** Probability of EFS for all patients with different FLT3/ITD AR. **D** Probability of OS for all patients with different FLT3/ITD AR. Abbreviations: AML acute myeloid leukemia, AR allelic ratio, EFS event-free survival, OS overall survival. EFS and OS were evaluated by the Kaplan–Meier method and compared by log-rank test. Source data are provided as a Source Data file.

FLT3/ITD AR ($P < 0.001$; Fig. 4A). Similarly, for chemotherapy-only pediatric AML patients, 5-year OS in patients with high AR was also significantly inferior to those harboring low AR (35.1% (21.5%) vs 54.5% (20.3%), $P = 0.015$; Fig. 4B). Within the cohort of SCT recipients, compared with high AR, low FLT3/ITD AR

had no effect on prognosis (5-year EFS: 46% (29.3%) vs. 65.6% (21.8%), $P = 0.18$; 5-year OS: 53.4% (27%) vs. 70.5% (20.1%), $P = 0.2$; Fig. 4C, D).

From further evaluation of the 139 childhood AML cases with low FLT3/ITD AR, survival analysis demonstrated that

SCT conferred no impact on prognosis when compared with chemotherapy-only recipients (5-year EFS: 46% (18.9%) vs. 39.8% (11.4%), $P = 0.18$; 5-year OS: 53.4% (27%) vs. 54.5% (20.3%), $P = 0.64$; Fig. 5A, B). However, when restricted to the high FLT3/ITD AR subgroup ($n = 161$), SCT had a favorable impact on 5-year EFS (65.6% (22.5%) vs. 14.5% (18.3%), $P < 0.001$; Fig. 5C) and 5-year OS (70.5% (20.1%) vs. 35.1% (21.5%), $P < 0.001$; Fig. 5D) compared with chemotherapy-only recipients.

**GO and early treatment response.** A total of 836 patients with de novo AML treated with AOGAAML03P1 and AAML0531 were included in the evaluation, and we identified 169 FLT3/ITD-positive patients whose clinical outcome data were included in further analyses. Of these 169 patients with FLT3/ITD mutation, 89 patients received standard chemotherapy (GO group) and were treated with AAML0531 ($n = 73$) and AAML03P1 ($n = 16$). The remaining 80 patients which were treated in the AAML0531 group ($n = 77$) and AAML03P1 ($n = 3$) received chemotherapy in the No-GO group only. When compared with the No-GO ARM, there were no significant differences in gender, CNSL, karyotype, WBC, PB blast, BM blast, and risk group in the GO group (Table S5).

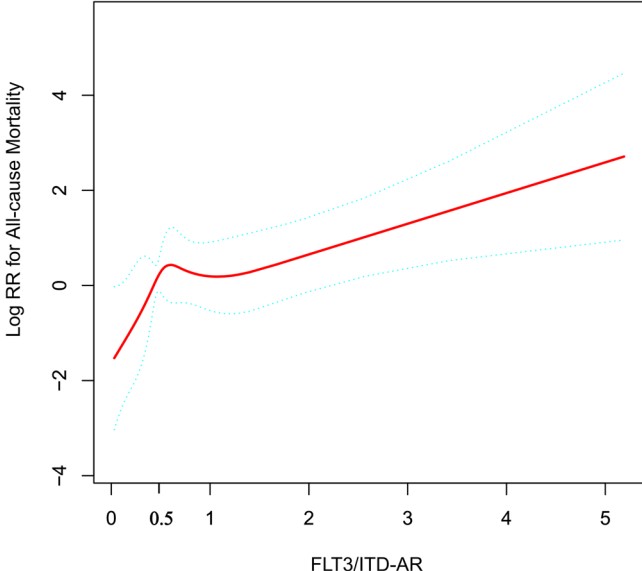

**Fig. 2 Restricted cubic spline of the association of FLT3/ITD AR with risk of all-cause mortality.** The resulting figures show the predicted log(relative risk) in the y-axis and the FLT3/ITD AR in the x-axis. Abbreviations: AR allelic ratio. The bold lines, upper boundaries and lower boundaries of notches represent the mean, max and min values. Data were expressed as means ± SD. Source data are provided as a Source Data file.

There were significantly more patients in FAB M2 in the GO vs. NO-GO cohort ($P = 0.042$). More children received GO therapy in AAML03P1 than in the NO-GO cohort (18% vs. 3.8%, $P = 0.003$). The median age of the No-GO ARM was higher than that of the GO cohort (12.1 years vs. 10.6 years, $P = 0.012$). No significant differences were found between high AR vs. low AR between the two treatment cohorts. Patients in the GO cohort accepted the initial dose of GO during induction course I, thus we evaluated the impact of GO on CR rates at the end of induction course. After the whole induction course, patients with GO had a CR rate of 80.5% vs. 73.4% for the No-GO cases ($P = 0.396$).

**Impact of GO and high AR FLT3/ITD.** Subsequently, we investigated the impact of GO in the high AR FLT3/ITD cohort. Among the high AR FLT3/ITD cohort, patients with GO had a similar CR rate of 68.9% vs. 67.5% for the No-GO group ($P = 0.854$) (Supplementary Table 6). Similarly, high AR patients did not experience significant benefit from GO alone in terms of 5-year EFS (27.1% (29.5%) vs. 31.4% (33.5%), $P = 0.67$; Fig. 6A) and OS (45% (28.7%) vs. 43% (31.6%), $P = 0.87$; Fig. 6B). However, our subgroup analysis demonstrated that the high AR FLT3/ITD patients who received consolidation chemotherapy plus GO ($n = 21$), had an inferior 5-year EFS (16.7% (37%) vs. 57.1% (55.1%), $P = 0.023$; Fig. 6C) and OS (25.6% (39.7%) vs. 71.4% (46.6%), $P = 0.014$; Fig. 6D) compared to those with SCT plus GO ($n = 14$).

**Multivariable analysis of event-free and overall survival among pediatric AML patients.** Multivariable Cox regression analysis, including FLT3/ITD AR, FLT3/ITD status, CEBPA status, NPM1 status, WT1 status, age, WBC, CNSL, karyotype, risk group, chemotherapy regimen, PB blast count, and BM blast count were performed to evaluate whether FLT3/ITD high AR was a favorable prognostic factor.

In result, high FLT3/ITD AR was shown to be an independent prognostic factor for poor EFS (HR = 4.8, 95% CI: 1.1–21.6, $P = 0.041$) and OS (HR = 3.6, 95% CI:1.1–12, $P = 0.034$), instead of the FLT/ITD mutation (EFS: HR = 0.9, 95% CI: 0.6–1.3, $P = 0.519$; OS: HR = 0.7, 95% CI: 0.5–1.1, $P = 0.116$) from the multivariable analysis. As expected, we showed that CEBPA and NPM1 mutations were also favorable independent prognostic factors for both EFS and OS, whereas WT1 was significantly associated with inferior EFS. Meanwhile, age (>10 years) exhibited a significantly negative influence on OS but not EFS. Conversely, WBC (>50 × 10$^9$/L) showed a strong association with an adverse EFS instead of OS. However, GO seemed to be not associated with EFS (HR = 0.9, 95% CI: 0.7–1.1, $P = 0.205$) or OS (HR = 1.0, 95% CI: 0.7–1.2, $P = 0.748$) (Table 4). In addition, PB blast, BM blast, risk group, chemotherapy regimen, karyotype, and risk group had no effect on EFS and OS.

**Table 2 Threshold effect analysis of the association between FLT3/ITD AR and risk of all-cause mortality using piece-wise linear regression.**

| FLT3/ITD AR | Crude | | Adjust[a] | |
|---|---|---|---|---|
| | Log*RR* (95% CI) | *P* value | Log*RR* (95% CI) | *P* value |
| <0.5 | 2.0 (0.9, 4.4) | 0.088 | 45.1 (0.9, 2281.6) | 0.057 |
| ≥0.5 | 1.0 (0.9, 1.1) | 0.715 | 1.6 (1.1, 2.3) | 0.013 |

Crude: none adjustment.
[a]Adjusted: adjustments were made for multiple comparisons, including Gender; Chemotherapy Protocol; Risk group; BM blast; Peripheral blood blasts; CNSL; FAB Category; Karyotype; WBC group; Age group; CEBPA status; WT1 status; NPM1 status.

**Table 3 Characteristics of patients with high or low FLT3/ITD AR.**

| Characteristics | Low AR (n = 152) | High AR (n = 189) | P value |
|---|---|---|---|
| Gender, n (%) | | | 0.071 |
| Male | 96 (63.2%) | 101 (53.4%) | |
| Female | 56 (36.8%) | 88 (46.6%) | |
| Age group (y) | | | 0.784 |
| <10 | 52 (34.2%) | 62 (32.8%) | |
| ≥10 | 100 (65.8%) | 127 (67.2%) | |
| FAB category | | | <0.001 |
| M0 | 0 (0.0%) | 2 (2.4%) | |
| M1 | 9 (12.5%) | 30 (35.7%) | |
| M2 | 27 (37.5%) | 15 (17.9%) | |
| M4 | 18 (25.0%) | 26 (31.0%) | |
| M5 | 15 (20.8%) | 11 (13.1%) | |
| M6 | 3 (4.2%) | 0 (0.0%) | |
| WBC group, n(%) | | | <0.001 |
| <50 × $10^9$/L | 76 (50.0%) | 60 (31.7%) | |
| ≥50 × $10^9$/L | 76 (50.0%) | 129 (68.3%) | |
| Karyotype | | | 0.141 |
| Normal | 74 (50.0%) | 99 (54.7%) | |
| inv(16) | 7 (4.7%) | 2 (1.1%) | |
| MLL | 8 (5.4%) | 6 (3.3%) | |
| t(8;21) | 7 (4.7%) | 4 (2.2%) | |
| Other | 52 (35.1%) | 70 (38.7%) | |
| Risk group, n (%) | | | <0.001 |
| Low risk | 44 (29.9%) | 0 (0.0%) | |
| Standard risk | 71 (48.3%) | 0 (0.0%) | |
| High risk | 32 (21.8%) | 189 (100.0%) | |
| CR status at end of course 1 | | | 0.175 |
| CR | 104 (69.8%) | 111 (60.3%) | |
| Not in CR | 42 (28.2%) | 69 (37.5%) | |
| Death | 3 (2.0%) | 4 (2.2%) | |
| CR status at end of course 2 | | | 0.048 |
| CR | 121 (84.0%) | 124 (72.5%) | |
| Not in CR | 20 (13.9%) | 42 (24.6%) | |
| Death | 3 (2.1%) | 5 (2.9%) | |
| SCT in 1st CR | | | 0.004 |
| No | 96 (69.1%) | 85 (52.8%) | |
| Yes | 43 (30.9%) | 76 (47.2%) | |
| Gemtuzumab ozogamicin treatment | | | 0.479 |
| No GO | 41 (49.4%) | 68 (54.4%) | |
| GO | 42 (50.6%) | 57 (45.6%) | |

The qualitative data were analyzed using the chi-square test and the quantitative data were compared using the Student's t-test (two-tailed).
*CR* complete remission, *SCT* stem cell transplantation, *GO* Gemtuzumab ozogamicin treatment.

## Discussion

The TARGET program is a COG-National Cancer Institute (NCI) collaboration that can be used to effectively study the pediatric cancer cohort[23]. The prevalence of FLT3/ITD mutation in the TARGET cohort was 18.4% (341 of 1857), which was consistent with previous pediatric reports[16]. Several prior reports[24–26] have demonstrated that FLT3/ITD mutation was significantly associated with higher WBC counts, higher PB blasts, higher BM blasts, older age, in addition to increasing probability to have FAB M1 phenotype and normal karyotype, where similar results could be found in this current study. Previous studies[27,28] also illustrated the presence of FLT3/ITD as an independent risk factor contributing to disease evolution and poor prognosis among AML patients. In this study cohort, we observed that after induction therapy, the CR rates in childhood AML that harbored the FLT3/ITD mutation were inferior to patients with FLT3/ITD wild-type, which indicated that mutant FLT3/ITD patients had a poor early treatment response. Furthermore, our survival analysis showed that FLT3/ITD mutation was identified for inferior EFS and OS rates compared with FLT3/ITD wild-type. Unfortunately, our multivariable analysis subsequently validated that FLT3/ITD mutation was not an independent prognostic factor. Taken together, the presence of FLT3/ITD had no significant effect on prognosis.

Based on our findings, we hypothesize that FLT3/ITD AR may play an important role in prognosis. Both the National Comprehensive Cancer Network (NCCN) and ELN guidelines incorporated FLT3/ITD mutation in the risk stratification of patients based on the AR with a threshold of 0.5[22,29]. However, the use of an AR threshold of 0.5, as described in the previous report, was only in the adult AML cohort. In regards to pediatric studies, the cut-off value of FLT3/ITD AR was only reported by two large-scale collaboration groups, which were the Children's Oncology Group (COG) with 630 childhood patients, and Japanese Pediatric Leukemia/Lymphoma Study Group (JPLSG) with a total of 369 samples. The former reported[16] that AR > 0.4 was a significant and independent prognostic factor for relapse in pediatric AML, while the latter reported[26] AR > 0.7 as the threshold for inferior EFS and OS. However, both of the studies were published several years ago and had much smaller sample sizes relative to the TARGET cohort (n = 1857), hence it was necessary to re-determine the optimal cut-off value of FLT3/ITD AR at present. Subsequently, we examined the relative risk for all-cause mortality of patients with different FLT3/ITD AR, which underwent a fitting curve analysis with the aim of confirming the optimal threshold of FLT3/ITD AR that was observed in such a large sample of FLT3/ITD-positive pediatric AML patients and finally the turning point was selected by using Cox proportional hazards models.

In this current study, those with high FLT3/ITD AR had a significantly higher WBC count, higher BM and PB blast, with inferior CR rates compared with those with low AR. Our survival analysis also showed that high FLT3/ITD AR had a strong impact on decreasing 5-year EFS and OS. From these results above, we suggested that the different FLT3/ITD allele ratios may represent an imbalance of FLT3/ITD alleles, ultimately reflecting the genetic heterogeneity and complexity of mutant FLT3/ITD AML[30]. Given the relationship between FLT3/ITD AR and prognosis, it has been reported that a high FLT3/ITD AR may suggest that the mutation occurred at an early stage and was a driving event for leukemogenesis, whereas a low AR suggests that the FLT3/ITD mutation was not a driving event for leukemia[31].

In the entire study, our analysis showed that SCT recipients of mutant FLT3/ITD could significantly improve EFS and OS when compared with those who did not receive SCT. These findings were consistent with the concept that SCT was indicated for AML patients with FLT3/ITD mutation[32,33]. When we focused on the impact of FLT3/ITD AR on the clinical outcome in pediatric AML, especially in the subgroup of chemotherapy-only patients, high FLT3/ITD AR was apparently associated with poorer EFS and OS when compared with low AR. In contrast, the 5-year EFS and OS of high FLT3/ITD AR could be significantly improved by SCT in our study. What's more important, after adjusting for a number of potential confounders, the results were still consistent in our K–M analysis. All of the abovementioned results suggested that SCT was more particularly indicated for pediatric AML patients with high AR instead of chemotherapy only.

As mentioned above, the prognosis of FLT3/ITD patients with chemotherapy-only is poor, especially those with high AR. While HCT increased the survival of patients in this cohort to approximately 65%, the use of alternative approaches such as GO is necessary to further improve outcomes. In this present study, we found that the addition of GO (3 mg/m²) to standard

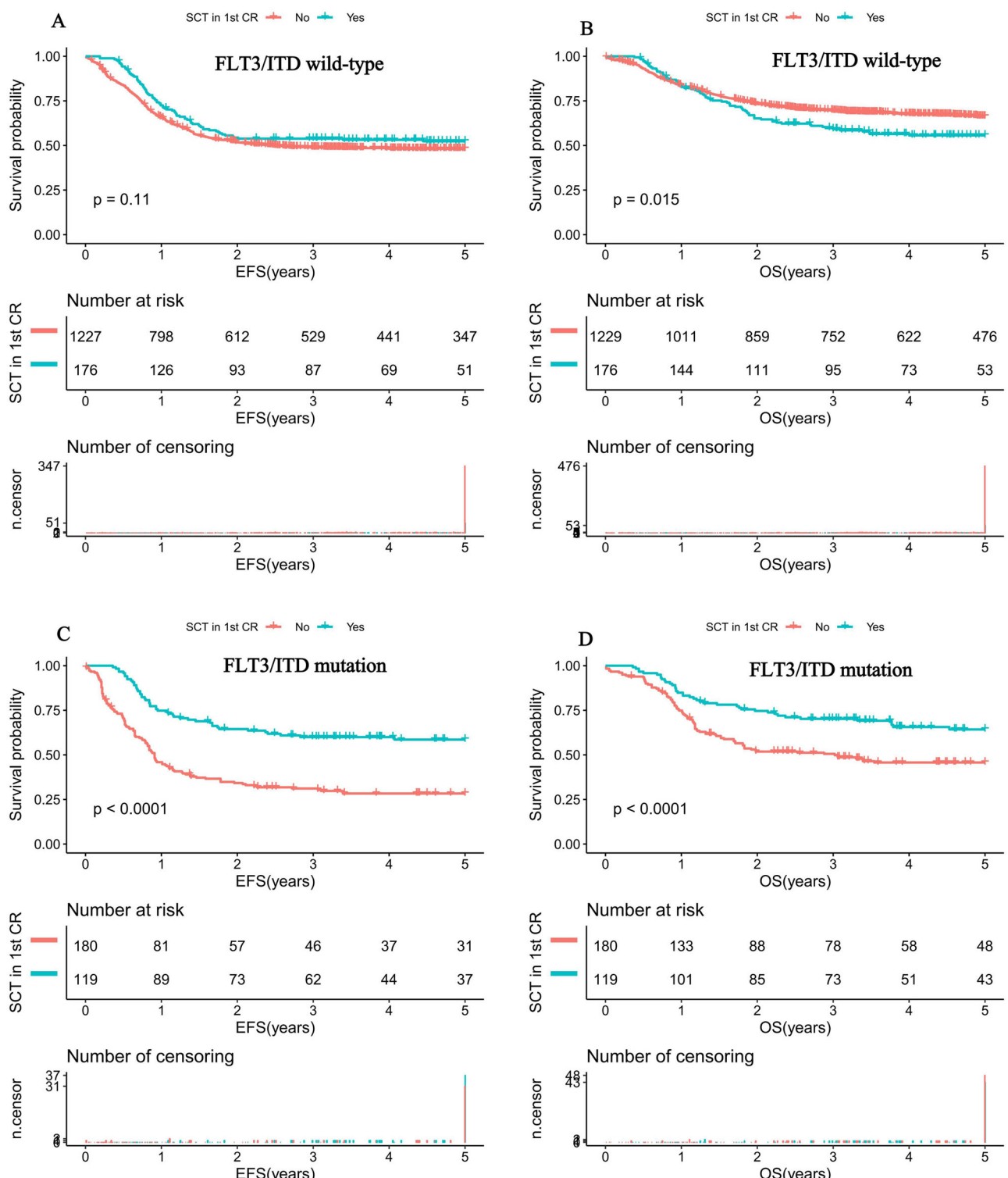

**Fig. 3 Survival curves of pediatric AML patients according to FLT3/ITD status and SCT status. A** Probability of EFS for patients with FLT3/ITD wild-type according to SCT status. **B** Probability of OS for patients with FLT3/ITD wild-type according to SCT status. **C** Probability of EFS for patients with FLT3/ITD mutation according to SCT status. **D** Probability of OS for patients with FLT3/ITD mutation according to SCT status. AML acute myeloid leukemia, AR allelic ratio, EFS event-free survival, OS overall survival. EFS and OS were evaluated by the Kaplan–Meier method and compared by log-rank test. Source data are provided as a Source Data file.

chemotherapy did not improve the initial CR rate. Subsequently, we assessed the outcomes for high AR FLT3/ITD patients with or without GO, and finally found no statistical differences on EFS and OS. However, through subgroup analysis, the prognosis of high AR patients who received SCT plus GO was significantly

better than those with chemotherapy plus GO, where the results were consistent with previous literature[34]. In total, our results showed that GO targeting with HCT consolidation might be a useful therapeutic strategy among high AR FLT3/ITD pediatric patients.

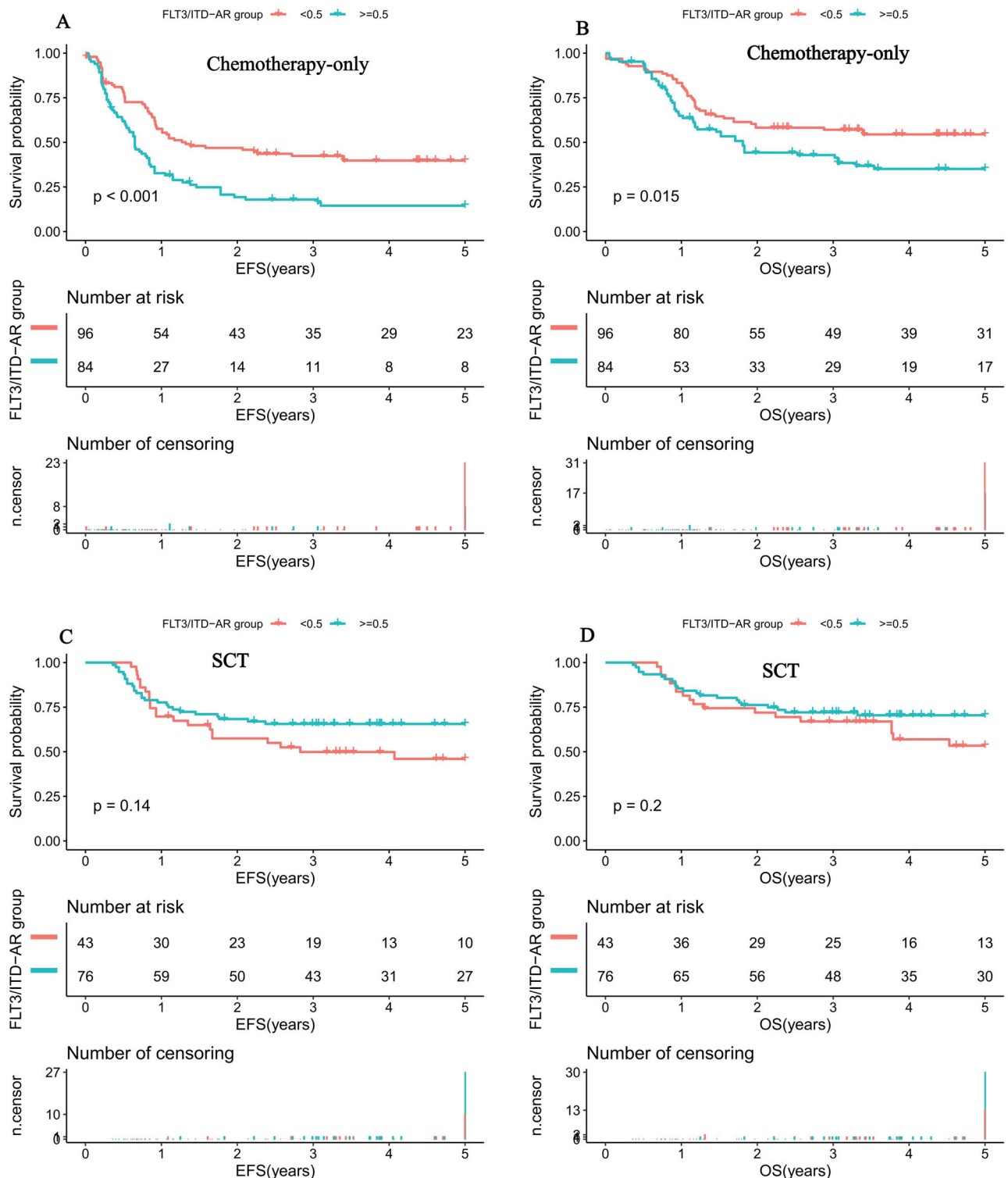

**Fig. 4 Survival curves of all pediatric AML patients according to the FLT3/ITD AR and SCT status. A** Probability of EFS for patients with chemotherapy-only in different FLT3/ITD AR. **B** Probability of OS for patients with chemotherapy-only in different FLT3/ITD AR. **C** Probability of EFS for patients with SCT in different FLT3/ITD AR. **D** Probability of OS for patients with SCT in different FLT3/ITD AR. AML acute myeloid leukemia, SCT stem cell transplantation, AR allelic ratio, EFS event-free survival, OS overall survival. EFS and OS were evaluated by the Kaplan–Meier method and compared by log-rank test. Source data are provided as a Source Data file.

In this retrospective analysis, the low FLT3/ITD AR patients with SCT did not seem to be superior to those without SCT. Given that patients with low AR had favorable prognoses with chemotherapy-only and no other data suggested any benefit from SCT, we recommend that patients with low FLT3/ITD AR should be prospectively stratified into chemotherapy-only therapy. Most importantly, these findings were later further established through multivariable analysis, which showed that high FLT3/ITD AR was an independent prognostic factor as expected.

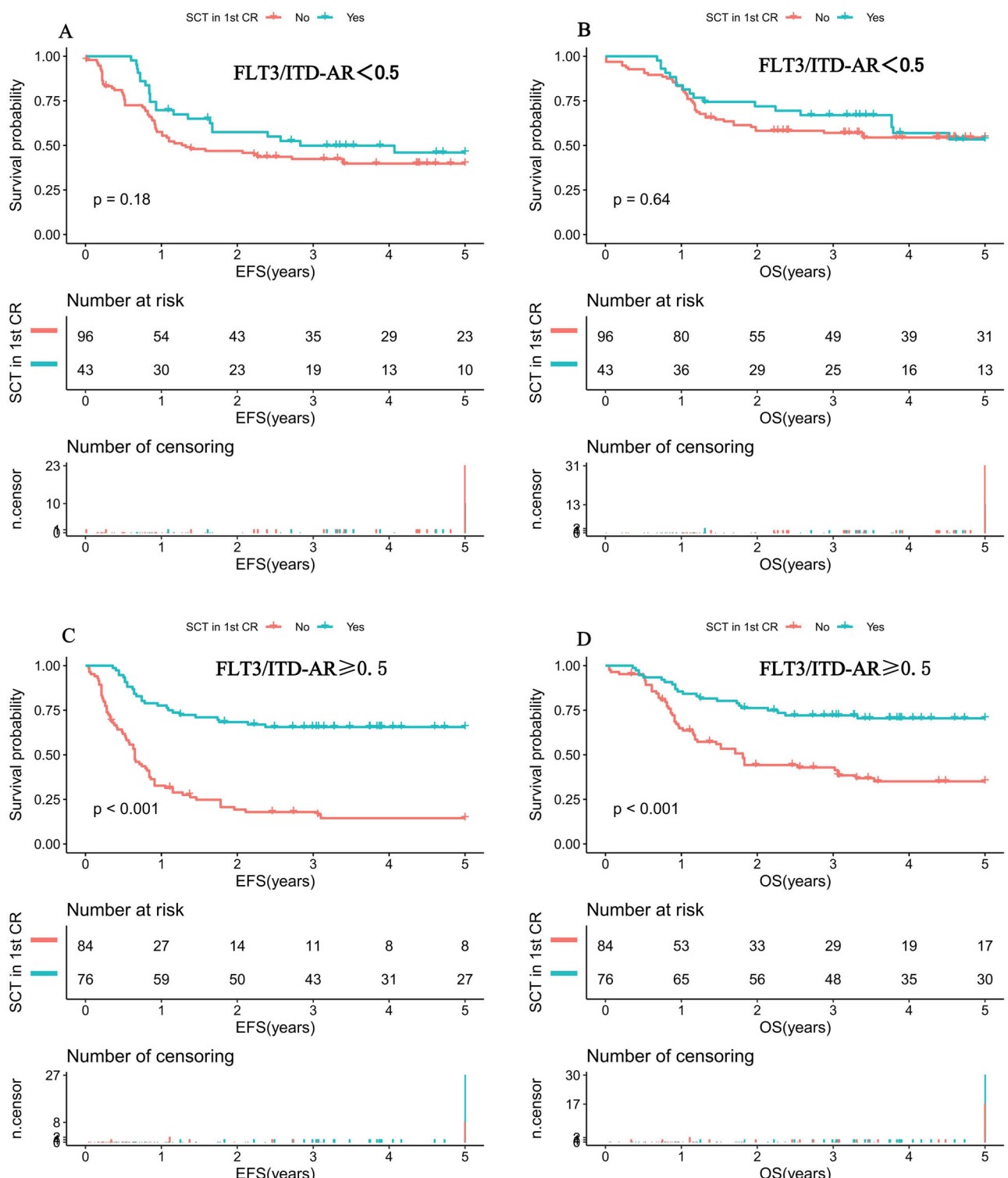

**Fig. 5 Survival curves of pediatric AML patients according to different FLT3/ITD AR and SCT status. A** Probability of EFS for patients with low FLT3/ITD AR according to SCT status. **B** Probability of OS for patients with low FLT3/ITD AR according to SCT status. **C** Probability of EFS for patients with high FLT3/ITD AR according to SCT status. **D** Probability of OS for patients with high FLT3/ITD AR according to SCT status. AML acute myeloid leukemia, SCT stem cell transplantation, CR complete remission, AR allelic ratio, EFS event-free survival, OS overall survival. EFS and OS were evaluated by the Kaplan–Meier method and compared by log-rank test. Source data are provided as a Source Data file.

In summary, our study demonstrated that the prevalence of FLT3/ITD mutation in the TARGET cohort was 18.4%, and our findings determined that FLT3/ITD AR of 0.5 is a clinically useful threshold for risk identification within the FLT3/ITD population. Furthermore, SCT may improve the outcome in childhood AML patients with high FLT3/ITD AR and may be further improved when combined with additional therapies such as GO.

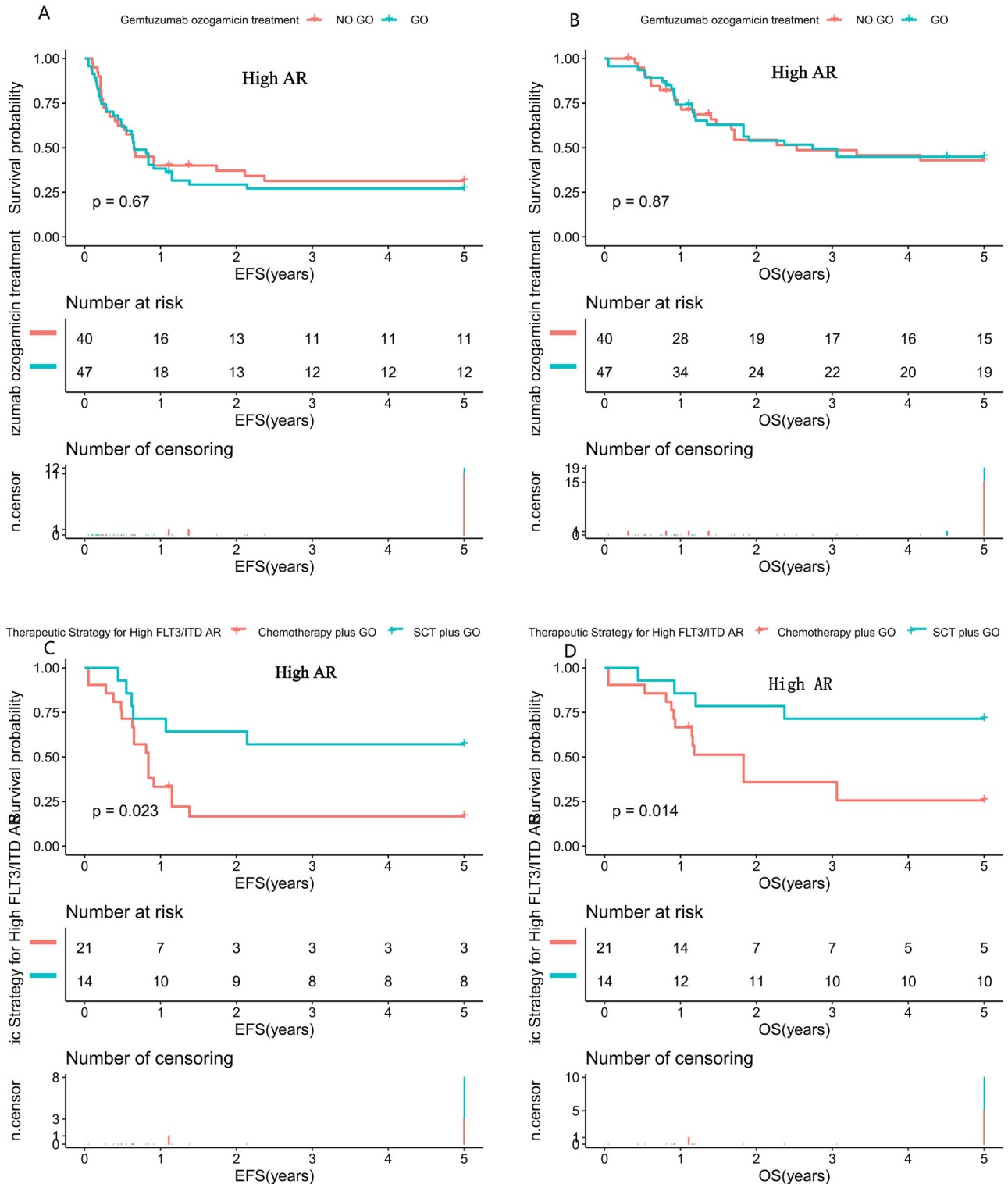

**Fig. 6 Survival curves of patients with high FLT3/ITD AR AML patients according to different treatment strategy. A** Probability of EFS for patients with and without GO treatment. **B** Probability of OS for patients with and without GO treatment. **C** Probability of EFS for patients with high FLT3/ITD AR according different treatment strategy. **D** Probability of OS for patients with high FLT3/ITD AR according to different treatment strategy. AML acute myeloid leukemia, SCT stem cell transplantation, GO Gemtuzumab Ozogamicin, AR allelic ratio, EFS event-free survival, OS overall survival. EFS and OS were evaluated by the Kaplan–Meier method and compared by log-rank test. Source data are provided as a Source Data file.

## Methods

**Study participants**. This study was approved by the ethics committee of National Cancer Institute's Office of Cancer Genomics. We obtained demographic, clinical characteristics, and laboratory data of AML in children under the age of 18 from the TARGET dataset (April 28, 2021). From September 1996 to December 2016, 2064 pediatric AML (non-M3) cases were enrolled in the TARGET database: 121 were excluded (secondary AML myeloid, $n = 35$; Down syndrome, $n = 86$), 86 were unable to be evaluated (loss of FLT3/ITD information, $n = 4$; insufficient data on therapy, $n = 82$), and finally, 1857 patients with childhood AML were enrolled in our research. The results published here

**Table 4 Multivariable analysis of prognostic factors for overall and event-free survival from study entry.**

| Variables | Event-free survival (EFS) | | Overall survival (OS) | |
|---|---|---|---|---|
| | HR (95% CI) | *P* value | HR (95% CI) | *P* value |
| High FLT3/ITD AR | 4.8 (1.1, 21.6) | 0.041 | 3.6 (1.1, 12) | 0.034 |
| FLT3/ITD mutation | 0.9 (0.6, 1.3) | 0.519 | 0.7 (0.5, 1.1) | 0.116 |
| CEBPA mutation | 0.2 (0.1, 0.5) | 0.001 | 0.2 (0.0, 0.6) | 0.008 |
| NPM1 mutation | 0.3 (0.1, 0.6) | <0.001 | 0.2 (0.1, 0.7) | 0.006 |
| WT1 mutation | 1.8 (1.3, 2.6) | 0.001 | 1.4 (0.9, 2.2) | 0.090 |
| Age > 10year | 1.2 (1.0, 1.5) | 0.084 | 1.5 (1.2, 2.0) | <0.001 |
| WBC > 50 × 10$^9$/L | 1.5 (1.2, 1.9) | <0.001 | 1.2 (0.9, 1.6) | 0.198 |
| CNSL | 1.2 (0.8, 1.8) | 0.418 | 1.2 (0.8, 2.0) | 0.353 |
| High-risk karyotype | 0.8 (0.6, 1.0) | 0.061 | 0.8 (0.5, 1.2) | 0.258 |
| GO | 0.9 (0.7, 1.1) | 0.205 | 1.0 (0.7, 1.2) | 0.748 |
| Chemotherapy protocol | 1.1 (0.8, 1.5) | 0.736 | 1.0 (0.7, 1.4) | 0.861 |
| High-risk group | 1.4 (0.6, 3.0) | 0.441 | 1.3 (0.4, 4.1) | 0.655 |
| BM blas t(%) | 1.0 (1.0, 1.0) | 0.272 | 1.0 (1.0, 1.0) | 0.784 |
| Peripheral blasts (%) | 1.0 (1.0, 1.0) | 0.482 | 1.0 (1.0, 1.0) | 0.549 |

The multivariable Cox proportional hazards model was used to determine the independent effect of prognostic factors on overall and event-free survival in AML patients.
*WBC* white blood cell counts, *CNSL* central nervous system leukemia, *CEBPA* CCAAT/enhancer binding protein alpha, *FLT3-ITD* fms-related tyrosine kinase 3, *NPM1* nucleophosmin 1, *WT1* wilms tumor 1, *GO* Gemtuzumab ozogamicin treatment.

were in whole or partly based upon data generated by the Therapeutically Applicable Research to Generate Effective Treatments (https://ocg.cancer.gov/programs/target) initiative, phs000218. Datasets from childhood patients with AML from 4 clinical trials were accessed through the TARGET database. These trials were registered as NCT00070174[35], NCT00372593[36], NCT01371981[37], and NCT00002798[38], respectively. The data used for this analysis are available at https://portal.gdc.cancer.gov/projects. The study was approved by the Ethics Committees of Office of Cancer Genomics (OCG). All the patient's guardians signed informed consent forms.

Patients were divided into three risk groups. For analysis purposes, low-risk patients were defined as those having inv(16)/t(16;16) or t(8;21), and high-risk patients were defined as those having monosomy 7 or del(5q). Patients with an accepted cytogenetic sample who did not meet the criteria for low or high risk were defined as standard risk. Stem cell transplantation (SCT) was performed in high-risk AML patients with first complete remission. Enrollment years were from 1996 to 2016, and the last follow-up years were 1997–2019, according to TARGET data.

**Therapeutic regimen.** The four therapeutic regimens included in this study differed in their basic composition of drugs. Tables S1 through S4 illustrate the four treatment regimens AAML03P1[35], AAML0531[36], AAML1031[37], and CCG-2961[38], respectively (Table S1 to Table S4). From the above-mentioned tables, we can see that only AAML03P1 and AAML0531 added Gemtuzumab Ozogamicin (GO) to intensive chemotherapy in pediatric AML. In brief, patients treated with AAML03P1 and AAML0531 were randomly assigned to one among two research arms, Arm A which included chemotherapy-alone (No-GO), and Arm B which included GO (3 mg/m$^2$) administered once during induction course I on Day 6, then once more during the intensification course II on Day 7.

**Statistical analysis.** The baseline characteristics represent the classification variables as frequency (%) and the median as continuous variables. Chi-square test was used for categorical variables and analysis of variance and the Kruskal–Wallis test was used for continuous variables. The multivariable Cox proportional hazards model was used to determine the independent effect of prognostic factors on overall and event-free survival in AML patients. In order to explore whether there was a non-linear relationship between FLT3/ITD AR and all-cause mortality, after adjusting for all covariates, the correlation fitting curve between FLT3/ITD AR and all-cause mortality was drawn using the restrictive cubic spline function following the Cox proportional hazards models. Threshold levels (i.e., turning points) were determined by trial and error, which involves selecting turning points along a predetermined interval and then chosing turning points. In order to assess the clinical outcome, we used the following concepts: complete remission (CR, was defined as less than 5% lymphoblasts in active hematopoietic bone marrow at the end of induction), event-free survival (EFS, defined as the start of the research to the timing of events for the first time, including induced failure, progress, or any cause of death and recurrence), and overall survival (OS, defined as the time between the beginning of learning and death from any cause). EFS and OS were evaluated by the Kaplan–Meier method and compared by log-rank test. All statistical analysis by SPSS statistical software version 22.0 and EmpowerStats (http://www.empowerstats.cn/). *P* < 0.05 was considered statistically significant.

**Reporting summary.** Further information on research design is available in the Nature Research Reporting Summary linked to this article.

## Data availability

The datasets used analyzed during the current study are publicly available from the AML dataset of TARGET dataset [https://target-data.nci.nih.gov/Public/AML/clinical/harmonized] or Figshare database [https://figshare.com/articles/dataset/Poor_Outcome_of_Pediatric_Patients_with_Acute_Myeloid_Leukemia_Harboring_High_FLT3_ITD_Allelic_Ratios/19809565]. The remaining data are available in the article and Supplementary Information. Source data are provided with this paper.

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

## Acknowledgements

The study was supported by the Guangdong Basic and Applied Basic Research Foundation (No. 2021A1515011809, DHZ), Guangzhou Science and Technology Program Key Projects (No. 201803010032, J.P.F.), Bethune Medical Scientific Research Fund Project (No. SCE111DS, J.P.F.), and Sun Yat-sen Sailing Scientific Research Project (YXQH202205, K.Y.Q.). We would like to thank the Office of Cancer Genomics and the help of Pamela C. Birriel.

## Author contributions

Conception and design of the study: K.Y.Q., X.Y.L., Y.L., J.P.F. and D.H.Z. Data collection: K.H. and Y.L. Data analysis and interpretation: K.Y.Q., X.Y.L., Y.L. Funding acquisition: J.P.F. and D.H.Z. J.P.F. and D.H.Z. contributed equally to this work as corresponding authors. Final approval of manuscript: All authors.

## Competing interests

The authors declare no competing interests.
