## [Peer Review File · Nature Communications]

Poor outcome of pediatric patients with acute myeloid leukemia harboring high FLT3/ITD allelic ratiosREVIEWER COMMENTS

Reviewer #1 (Remarks to the Author): expert in AML and mutation studies

The authors have made an excellent attempt to assess the role of FLT3 ITD MT and AR in 1857 pediatric AML patients from the TARGET data set.

They show that 18.4% of patients had a FLT3 ITD and that this was associated with decreased CR rates, decreased EFS, and decreased OS. They further demonstrated that FLT3 ITD AR >0.5 was prognostic and in comparison to AR of <0.5, had a better impact on prognosticating for CR rates, EFS and OS.

They also show that the benefit for HSCT was in the AML patients with FLT3 ITD AR >0.5.

While this is a good study it doesn't add too much to the existing literature and doesn't have important granular data with regards to individual treatment regimens, clinical trial usage of FLT3 inhibitors, comorbidities, and has a limited set of mutations to assess in the MVA.

As such i don't think this paper meets the mark for Nature Communications.

Reviewer #2 (Remarks to the Author): expertise in paediatric AML

Qiu et al. first reported prognostic significance of FLT3/ITD allelic ratio(AR) among 1857 pediatric AML from TARGET dataset.

Although this is the first report among pediatric AML, but significance of FLT3/ITD ratio already reported in adult AML. Another critical point of this paper is the description that SCT may improve the outcome patients with high AR. These dataset include different clinical studies (AAML03P1, AAML0531, AAML1031) and indications of SCT were different among these clinical studies. So, Figure 4 should be reanalyse in each clinical studies.

Reviewer #3 (Remarks to the Author): expertise in survival analyses biostatistics

This is an interesting study on the prognostic value of FLT3/ITD allelic ratios (AR) in pediatric acute myeloid leukemia.

Unfortunately, the statistics used in this study needs to be revised as they can have a large impact on the outcomes.

English language needs also to be improved.

Methods

Why did the authors not exclude the APL patients, while these patients have a good prognosis and receive a different therapy.

2.2 In the first sentence 'mean' is mentioned as I think the authors mean 'median' as continuous variables are hardly normally distributed. Moreover, you can only show the standard deviation between brackets and never +/-, SDs are not equal to a confidence interval.

Results

3.1 The authors conclude that there is no gender difference between the wild type and mutation group, while in Table 1 the p-value=0.029. This is a significant difference which should be mentioned in the results.

3.2

a. It would be valuable to add the median survival time of both groups.

b. Did the authors take into account that from the recent patient population you will have <5 years of follow-up?

c. It is unclear how the p-values of 5-year EFS/OS were calculated. Are these p-values based on Figure 1A and 1B? While in this figures the authors show survival times of 10 years and to

calculate the right p-value using log-rank you need to truncate your follow-up time at 5 years.
d. 5-year EFS and OS are mentioned as survival rate % +/- undefined value. Do the authors mean the SD? Please see my comment at 2.2. Please give the standard error between brackets or calculate the 95% confidence interval.

3.3 Did the authors check for interaction terms between variables that they have added in the protocol? For example 'risk group' is related to the 'FAB category'. It could also be that gender is interacting with some of the disease factors.

3.5 The authors compare only SCT and chemotherapy. Did they took immortal time bias into account, because this bias could result in better outcomes of the SCT group.

Point-by-point response to the reviewers' comments

Reviewer #1 (Remarks to the Author): expert in AML and mutation studies

>

>The authors have made an excellent attempt to assess the role of FLT3 ITD MT and AR in 1857 pediatric AML patients from the TARGET data set. They show that 18.4% of patients had a FLT3 ITD and that this was associated with decreased CR rates, decreased EFS, and decreased OS. They further demonstrated that FLT3 ITD AR >0.5 was prognostic and in comparison to AR of <0.5, had a better impact on prognosticating for CR rates, EFS and OS. They also show that the benefit for HSCT was in the AML patients with FLT3 ITD AR >0.5.

>

>While this is a good study it doesn't add too much to the existing literature and doesn't have important granular data with regards to individual treatment regimens, clinical trial usage of FLT3 inhibitors, comorbidities, and has a limited set of mutations to assess in the MVA.

>**Response:** We thank the reviewer for the kind comments. Now I want to introduce something especial in my manuscript. Firstly, previous reports on the threshold of childhood FLT3/ITD AR are very few, and they are all small sample reports. However, we determine the optimal cut-off value for FLT3/ITD AR is 0.5 among pediatric AML from the largest-scale sample, which could serve as an useful marker for risk-stratified therapy and prognostic evaluation. Secondly, we confirm that high FLT3/ITD AR, but not the presence of FLT3/ITD, are independent poor prognostic factors in pediatric AML. Furthermore, low FLT3/ITD AR might not require stem cell transplantation, while high AR did. Lastly, to the best of our knowledge, this is the first study using generalized additive models and threshold effect analysis to identify the threshold of AR, which makes the conclusion more credible. Base on those reason above, I think my manuscript is still very meaningful and practical.

As to individual treatment regimens, clinical trial usage of FLT3 inhibitors and comorbidities and so on, because there is no content about these in the things I downloaded from the TARGET database. Because the reviewer mentioned this information, I also want to add what the reviewer said to my manuscript. So I contacted the PI of TARGET, but they couldn't provide enough information (see the email screenshot below) either. Therefore, it's a pity that I can't meet your requirements on these contents and I have tried my best. However, due to your reminder on treatment, I found that Gemtuzumab ozogamicin treatment is of special significance in childhood AML with FLT3/ITD, so we added section 3.6 and Figure 7 in my revised manuscript.

Section 3.6 is that "Then we investigated the impact of GO in high AR FLT3/ITD cohort. Among the 125 patients with high AR FLT3/ITD, 57 childhood ALL had accepted induction GO and 68 pediatric patients were treated on the No-GO arm. As a result, high AR FLT3/ITD patients didn't experience significant benefit from GO alone in terms of 5-year EFS (29.41%(10%) vs 23.92%(8%), $P=0.663$; **Figure 7A**) and OS (43.33%(11%) vs 37.66%(10%), $P=0.425$; **Figure 7B**). Furtherly,

our subgroup analysis demonstrated that the high AR FLT3/ITD patients who received consolidation chemotherapy plus GO (n=12), had an inferior 5-years EFS (8.33%(2%) vs 61.11%(5%), $P < 0.001$; **Figure 7C**) and OS (20.83%(8.2%) vs 72.22%(7.5%), $P < 0.001$; **Figure 7D**) to those with SCT plus GO (n=18)". From section 3.6, we draw the conclusion that SCT may improve the outcome in childhood AML patients with high FLT3/ITD AR, and maybe further improved when combined with additional therapies such GO.

I hope that the changes I have made resolve all your concerns about the article. I'm more than happy to make any further changes that will improve the paper and facilitate successful publication.

 邮件可翻译为中文 立即翻译

There were no FLT3 inhibitors used in these patients.

Best,
Soheil

Soheil Meshinchi, MD, Ph.D.
smeshinc@fredhutch.org

Excuse the brevity of this message. Sent from my hand held device.

On Jan 26, 2022, at 3:55 PM, 邱 <398067206@qq.com> wrote:

Dear Soheil Meshinchi,

I would like to use all the data on children with acute myeloid leukemia with FLT-ITD mutation in TARGET dataset. But Now I found no information about FLT3 inhibitors and comorbidities among acute myeloid leukemia, which was not showed in the TARGET dataset. Could you provide these information for me? I am looking forward to your reply.

Sincerely

Yun-xin Qiu

Reviewer #2 (Remarks to the Author): expertise in paediatric AML

>Qiu et al. first reported prognostic significance of FLT3/ITD allelic ratio(AR) among 1857 pediatric AML from TARGET dataset.

>Although this is the first report among pediatric AML, but significance of FLT3/ITD ratio already reported in adult AML. Another critical point of this paper is the description that SCT may improve the outcome patients with high AR. These dataset include different clinical studies (AAML03P1, AAML0531, AAML1031) and indications of SCT were different among these clinical studies. So, Figure 4 should be reanalyse in each clinical studies.

Response: Thanks for your useful suggestion. We acknowledge your comments very much, which are valuable in improving the quality of our manuscript. Firstly, although the significance of FLT3/ITD ratio already reported in adult AML, the FLT3/ITD ratio among children remains uncertain. Secondly, it is important to determine a clinically useful threshold of AR for risk identification within the FLT3/ITD population among children. Thirdly, although we made the thresholds consistent with those of adults, we did not calculate them by referring to adult thresholds, it just happens to be consistent with the threshold of adults. We through generalized additive models and threshold effect analysis to identify the threshold of AR, which makes the conclusion more credible, and this method was not used in

previous literature. Totally, I think it is necessary for us to make this study to confirm the threshold of AR especially in pediatric AML.

In addition, we agree with the reviewer about that “Figure 4 should be reanalyse in each clinical studies.” However, Figure 5 demonstrate SCT may improve the outcome patients with high AR indeed, we think the reviewer mean Figure 5 instead Figure 4. So we perform K-M survival analysis by adjust variable “chemotherapy” and other variables and get Figure 6 in our revised manuscript. Though Figure 6, we confirm SCT still improve the outcome patients with high AR although among these different clinical studies.

>**Reviewer #3** (Remarks to the Author): expertise in survival analyses biostatistics

>This is an interesting study on the prognostic value of FLT3/ITD allelic ratios (AR) in pediatric acute myeloid leukemia. Unfortunately, the statistics used in this study needs to be revised as they can have a large impact on the outcomes.

>English language needs also to be improved.

>**Response:** Thanks for your suggestions. We feel sorry for our poor writings, however, we do invite a friend of us who is a native English speaker from USA help polish our article. Due to our friend’s help, the article was edited extensively. And we hope the revised manuscript could be acceptable for you.

>Methods

>Why did the authors not exclude the APL patients, while these patients have a good prognosis and receive a different therapy.

Response: I'm sorry I didn't describe this clearly in the manuscript. In fact, APL is excluded in my manuscript. The 2064 patients we downloaded from the target database are non-M3 patients. Now I correct it in the section of Methods like this “From September 1996 to December 2016, 2,064 childhood AML (**non-M3**) cases were enrolled in the TARGET database” in my manuscript.

>

>2.2 In the first sentence 'mean' is mentioned as I think the authors mean 'median' as continuous variables are hardly normally distributed. Moreover, you can only show the standard deviation between brackets and never +/-, SDs are not equal to a confidence interval.

>**Response:** Thanks for your professional suggestion. Your comments are of great importance to our article. Actually, the first sentence 'mean' is indeed “median” as you say. I correct my mistake in my revised manuscript in section 2.2.

>Results

>3.1 The authors conclude that there is no gender difference between the wild type and mutation group, while in Table 1 the p-value=0.029. This is a significant difference which should be mentioned in the results.

>**Response:** Thanks for your help. We feel really sorry for our carelessness. As you say, there is significantly gender difference between the wild type and mutation group,

so we add “Mutant FLT3/ITD were more common in male when compared with FLT3/ITD wild-type in terms of gender distribution (57.8% vs 51.3%, $P=0.029$)” in our section 3.1.

>3.2

>a. It would be valuable to add the median survival time of both groups.

Response: Thank you again for your professional advice. We can really benefit a lot with your help. We add “The median follow-up time (and range) for childhood AML alive at last contact 2.9 (0.1-10.9) years for patients with mutant FLT3/ITD and 3.6 (0.1-11.3) years for those with wild-type” in our revised manuscript in section 3.2.

>b. Did the authors take into account that from the recent patient population you will have <5 years of follow-up?

Response: Thanks your question. Of course, a small number of patients will be followed up for less than 5 years, but patients with malignant diseases such as leukemia need 5 years to determine whether they relapse or die. Generally, we think that if leukemia patients still survive for more than 5 years, it can be considered that patients will probably not have recurrence and other events, so we use K-M to estimate the 5-year survival rate.

>c. It is unclear how the p-values of 5-year EFS/OS were calculated. Are these p-values based on Figure 1A and 1B? While in this figures the authors show survival times of 10 years and to calculate the right p-value using log-rank you need to truncate your follow-up time at 5 years.

Response: Once again, we acknowledge your comments very much, which are valuable in improving the quality of our manuscript. Actually, the p-value should be calculated at 5 years, so we truncate our follow-up time at 5 years and make a new Figure 1 and get some new data in my revised manuscript.

>d. 5-year EFS and OS are mentioned as survival rate % +/- undefined value. Do the authors mean the SD? Please see my comment at 2.2. Please give the standard error between brackets or calculate the 95% confidence interval.

Response: We feel sorry that we did not provide enough information about EFS and OS. In fact, I mean the SD. Now I correct my mistake and give the standard error between brackets in my revised manuscript.

>

>3.3 Did the authors check for interaction terms between variables that they have added in the protocol? For example 'risk group' is related to the 'FAB category'. It could also be that gender is interacting with some of the disease factors.

>**Response:** Thanks for your comments. In fact, we perform generalized additive models after adjusting for these possible factors related to adverse clinical outcomes, including gender, chemotherapy protocol; risk group; BM blast; PB blasts; CNSL; FAB category; karyotype; WBC group; age group; CEBPA status; WT1 status; NPM1 status, so we already check for interaction terms between variables that added in the protocol.

>3.5 The authors compare only SCT and chemotherapy. Did they take immortal time bias into account, because this bias could result in better outcomes of the SCT group.

Response:We thank the reviewer for pointing this out. We really should take immortal time bias into account, so we perform K-M survival analysis again by adjust “years of diagnosis, Gender; Chemotherapy Protocol; Risk group; BM blast; Peripheral blood blasts; CNSL; FAB Category; Karyotype; WBC group; Age group; CEBPA status; WT1 status; NPM1 status”, and we find SCT group have a better EFS and OS than chemotherapy group in AR \geq 0.5 FLT3/ITD(Figure 6). You can I add these”After adjusting for potential confounders such as “years of diagnosis, Gender; Chemotherapy Protocol; Risk group; BM blast; Peripheral blood blasts; CNSL; FAB Category; Karyotype; WBC group; Age group; CEBPA status; WT1 status; NPM1 status”, our K-M survival analysis further confirmed that for patients with low FLT3/ITD AR, SCT had no effect on prognosis when compared with those with chemotherapy-only in terms of EFS ($P=0.2378$, **Figure 6A**) and OS ($P=0.5224$, **Figure 6B**). In addition, SCT still showed a significantly beneficial prognosis on 10-year EFS ($P<0.001$; **Figures 6C**) and 10-year OS ($P<0.001$; **Figures 6D**)” in my reviewed manuscript in section 3.5.

REVIEWER COMMENTS

Reviewer #1 (Remarks to the Author): expertise in mutations in AML

The authors have made a suitable attempt to justify including their paper in the Journal. While I understand that the FLT3 allelic ratio story is not novel in adults, this data is important for pediatric patients. The FLT3 allelic ratio cut off and the role of allogeneic stem cell transplant are suitably highlighted.

That being said, it is indeed disappointing that there is no granular data on the pediatric AML protocols included in the study. This is particularly important given that the authors are now commenting on the role of gemtuzumab in FLT3 mutated AML and even in the transplant arm. This drug has been used in different dosing strategies. It is not sufficient just to mention the name of the drug and its impact on survival. All the trials that have been included in project TARGET are available via clinicaltrials.gov.

For the sake of clarity, I urged the authors to make a better attempt at generating supplementary data to clarify treatment strategies. If they cannot do this, they should delete the section on GO. In addition to this, the English language is currently not suitable and editorial assistance should be provided.

Reviewer #4 (Remarks to the Author): expertise in survival analysis and biostatistics

This manuscript addresses a compelling clinical challenge regarding FLT3-ITD mutations and allelic ratios as prognostic factors in pediatric AML patients. The use of the TARGET dataset provides a sufficient resource to address the authors' hypothesis. Determination of a FLT3-ITD allelic ratio cut-off for pediatric patients serves as a potentially impactful precision medicine marker that may be useful to inform patient prognosis and guide treatment decisions. While the study goals are meaningful and interesting, there are some major concerns with the manuscript as currently submitted.

Major Comments

1. Major grammatical revision required. There are numerous sentences throughout that are challenging to interpret as written, as well as substantial grammatical errors.
2. The methods used for determining the threshold for dividing FLT3 allelic ratio groups need to be revised. First, the endpoint used for this analysis is unclear. The manuscript states "adverse clinical outcomes" as the endpoint, which in the methods are defined as "induction failure, progression or death and relapse". Based on this, it seems as though the endpoint is a categorical variable (yes/no), as opposed to continuous. As such, logistic regression is warranted rather than linear regression. Clarification of the endpoint should be provided, and the appropriate regression method should be implemented.
3. Results, 3.7 Multivariate analysis and Table 4: There are important predictor variables that should be added into these models, such as risk group, chemotherapy regimen, blast count, etc. Without including other factors such as these that are known to be associated with outcomes, you cannot accurately quantify the impact of FLT3.
4. Discussion: This statement is unclear and does not seem to match the data presented – "FLT3/ITD mutations might have a poor prognosis, but the presence of FLT3/ITD had no significant effect on the prognosis."
5. Based on previous reviewers' comments and the authors' responses, the survival time for this study is still unclear (5- versus 10-year). The authors' responses are inconsistent within themselves. This needs to be better addressed.

Minor Comments

1. What classification system was used to define the risk groups?
2. Please provide a p-value for this sentence (Results, 3.2 Clinical outcome...): "The median follow-up time (and range) for childhood AML alive at last contact 2.9 (0.1-10.9)..."
3. Results, 3.5 The impact of SCT...– remove " from text: After adjusting for potential confounders such as "years of diagnosis, Gender; Chemotherapy Protocol...", our K-M survival [...]
4. Results, 3.7 Multivariate analysis and Table 4: This is multivariable analysis, not multivariate (because it includes one outcome at a time with multiple predictor variables). Please correct.

5. Discussion: This statement over-states the robustness of results from this single study and should be modified to deemphasize the definitiveness of the allelic ratio threshold determined herein – "...this is the first time to use such a method to identify the threshold of AR, which makes the threshold credible and reliable."

a. Same for this statement – "Interestingly, as a result, FLT3/ITD AR of 0.5 was confirmed as the prognostic threshold..."

6. Figure 6 should be removed and only Table 4 included to accurately report these results.

Point-by-point response to the reviewers' comments

Reviewer #1 (Remarks to the Author): expert in AML and mutation studies

The authors have made a suitable attempt to justify including their paper in the Journal. While I understand that the FLT3 allelic ratio story is not novel in adults, this data is important for pediatric patients. The FLT3 allelic ratio cut off and the role of allogeneic stem cell transplant are suitably highlighted.

Response: We thank the reviewer for this insightful comment. We fully agree with the reviewer that “FLT3 allelic ratio story is not novel in adults” as many publications have stated that a crucial cut-off value of 0.5 for FLT3 allelic ratio has been well established in adults and confirmed in European LeukemiaNet (ELN 2017)

However, as the reviewer correctly stated, this data is important for pediatric patients and few publication reported before. More importantly, our primary purpose is to determine an optimal cut-off value for FLT3-ITD AR in a large-scale study of 1,857 pediatric AML from the Therapeutically Applicable Research to Generate Effective Treatments (TARGET) database, in order to provide strong evidence for risk-stratified therapy and prognostic evaluation in the FLT3/ITD positive pediatric patient population. Furthermore, in our study, we demonstrated that the optimal cut-off value for FLT3/ITD AR is 0.5 in pediatric AML and stem cell transplant may improve the outcome in childhood AML patients with high FLT3/ITD AR.

2. That being said, it is indeed disappointing that there is no granular data on the pediatric AML protocols included in the study. This is particularly important given that the authors are now commenting on the role of gemtuzumab in FLT3 mutated AML and even in the transplant arm. This drug has been used in different dosing strategies. It is not sufficient just to mention the name of the drug and its impact on

survival. All the trials that have been included in project TARGET are available via clinicaltrials.gov.

***Response:** Once again, we are grateful for such insightful comments. On the reviewer's first point, we fully agree with the reviewer that granular data on the pediatric AML protocols should be included in the study, and we apologized for our overlook on this regard. As the reviewer pointed out, since all trials that are included in project TARGET are available via clinicaltrials.gov, we have since included all granular data on the pediatric AML protocols, now supplemented in **Table S1 to Table S4, Supplementary Information**. Moreover, we have now added add section "2.2 Therapeutic regimen" in our revised manuscript and introduce the four regimen and the dose of GO in details (please see revised manuscript, Section 2.2, page 4).*

*We also agree that it is indeed insufficient just to mention the drug name and its impact on survival. In respond to this comment, we add the current section "**3.6 GO and early treatment response**" to report that baseline characteristics of FLT3/ITD positive pediatric AML in the GO and No-GO cohorts and their respective CR rates. These additional results can be found in **Table S5 and S6** in the *Supplementary Information*. In addition, "**Section 3.7 Impact of GO and high AR FLT3/ITD**" also states impact of GO on high AR survival.*

For the sake of clarity, I urged the authors to make a better attempt at generating supplementary data to clarify treatment strategies. If they cannot do this, they should delete the section on GO.

***Response:** We thank the reviewer for this insightful suggestion. Per the reviewer's suggestion, we have generated relevant data to clarify the treatment strategies (please see revised manuscript, **Table S1 to Table S6, Supplementary Information, page 28 to page 34**).*

In addition to this, the English language is currently not suitable and editorial assistance should be provided.

Response: We apologize for the poor English commands. We have sought a professional IACET accredited medical editor to proofread this manuscript. We hope that this manuscript could be deemed satisfactory in terms of language proficiency.

Reviewer #4 (Remarks to the Author): expertise in survival analysis and biostatistics

This manuscript addresses a compelling clinical challenge regarding FLT3-ITD mutations and allelic ratios as prognostic factors in pediatric AML patients. The use of the TARGET dataset provides a sufficient resource to address the authors' hypothesis. Determination of a FLT3-ITD allelic ratio cut-off for pediatric patients serves as a potentially impactful precision medicine marker that may be useful to inform patient prognosis and guide treatment decisions. While the study goals are meaningful and interesting, there are some major concerns with the manuscript as currently submitted.

Major Comments

1. Major grammatical revision required. There are numerous sentences throughout that are challenging to interpret as written, as well as substantial grammatical errors.

Response: We apologize for the poor English commands. We have sought a professional IACET accredited medical editor to proofread this manuscript. We hope that this manuscript could be deemed satisfactory in terms of language proficiency.

2. The methods used for determining the threshold for dividing FLT3 allelic ratio groups need to be revised. First, the endpoint used for this analysis is unclear. The manuscript states "adverse clinical outcomes" as the endpoint, which in the methods are defined as "induction failure, progression or death and relapse". Based on this, it seems as though the endpoint is a categorical variable (yes/no), as opposed to continuous. As such, logistic regression is warranted rather than linear regression. Clarification of the endpoint should be provided, and the appropriate regression method should be implemented.

Response: We apologize for the unclear description that leads to the confusion of the reviewer. For the reviewer's information, this adverse event refers to all-cause

mortality. In fact, we are neither using logistic regression, nor linear regression, we are using Cox regression. We have placed “the correlation fitting curve between FLT3/ITD AR and all-cause mortality was drawn by using the restrictive cubic spline function following on Cox proportional hazards models” into section 2.3 (please see revised manuscript, page 5). Furthermore, we have changed these in the corresponding results (Section 3.3, page 7 to page 8) and discussion (page 12 to page 13).

3. Results, 3.7 Multivariate analysis and Table 4: There are important predictor variables that should be added into these models, such as risk group, chemotherapy regimen, blast count, etc. Without including other factors such as these that are known to be associated with outcomes, you cannot accurately quantify the impact of FLT3.

Response: *We thank the reviewer for pointing this out this shortcoming. Per the reviewer’s suggestion, we have added “risk group, chemotherapy regimen, PB blast count and BM blast count” into these models in our revised manuscript (Please see Table 4 and Section 3.7).*

4. Discussion: This statement is unclear and does not seem to match the data presented – “FLT3/ITD mutations might have a poor prognosis, but the presence of FLT3/ITD had no significant effect on the prognosis.”

Response: *We apologize for the misleading representation of our data. Upon consideration, we conclude that this sentence “FLT3/ITD mutations might have a poor prognosis” is incorrect We have since deleted this sentence from the revised manuscript.*

5. Based on previous reviewers’ comments and the authors’ responses, the survival time for this study is still unclear (5- versus 10-year). The authors’ responses are inconsistent within themselves. This needs to be better addressed.

Response: *We apologize for the unclear presentation of our data. For the reviewer's information, we have revised our data for clearer presentation. As seen in the revised manuscript, we have depicted the 5 years of survival in all figures (Figure 3 to Figure 6, revised manuscript, page 24 to page 27). We have also included this figure below for your perusal:*

Figure 3 Survival curves of pediatric AML patients according to FLT3/ITD status and SCT status. (A) Probability of EFS for patients with FLT3/ITD wild-type according to SCT status. (B) Probability of OS for patients with FLT3/ITD wild-type according to SCT status. (C) Probability of EFS for patients with FLT3/ITD mutation according to SCT status. (D) Probability of OS for patients with FLT3/ITD mutation according to SCT status.

Figure 4 Survival curves of all pediatric AML patients according to the FLT3/ITD AR and SCT status. (A) Probability of EFS for patients with chemotherapy-only in different FLT3/ITD AR. (B) Probability of OS for patients with chemotherapy-only in different FLT3/ITD AR. (C) Probability of EFS for patients with SCT in different FLT3/ITD AR. (D) Probability of OS for patients with SCT in different FLT3/ITD AR.

Figure 5 Survival curves of pediatric AML patients according to different FLT3/ITD AR and SCT status. (A) Probability of EFS for patients with low FLT3/ITD AR according to SCT status. (B) Probability of OS for patients with low FLT3/ITD AR according to SCT status. (C) Probability of EFS for patients with high FLT3/ITD AR according to SCT status. (D) Probability of OS for patients with high FLT3/ITD AR according to SCT status.

Figure 6 Survival curves of patients with high FLT3/ITD-AR AML patients according to different treatment strategy. (A) Probability of EFS for patients with and without GO treatment. (B) Probability of OS for patients with and without GO treatment. (C) Probability of EFS for patients with high FLT3/ITD AR according different treatment strategy. (D) Probability of OS for patients with high FLT3/ITD AR according to different treatment strategy.

Minor Comments

1. What classification system was used to define the risk groups?

Response: *We apologize for the lack of description on the risk classification clearly. For better description of the risk groups, we have added the following sentence in the revised manuscript (section 2.1 Study participant). These new additions are listed below.*

“Low-risk patients were defined for analysis purposes as those having $inv(16)/t(16;16)$ or $t(8;21)$, and high-risk patients were defined as those having monosomy 7 or $del(5q)$. Patients with an accepted cytogenetic sample who did not meet criteria for low or high risk were defined as standard risk”

2. Please provide a p-value for this sentence (Results, 3.2 Clinical outcome...): “The median follow-up time (and range) for childhood AML alive at last contact 2.9 (0.1-10.9)...”

Response: *Per the reviewer’s suggestion, we have provided a p-value for this sentence in our revised manuscript. The new sentence is listed below.*

*“The median follow-up time (and range) for childhood AML alive at last contact 2.9 (0.1-10.9) years for patients with mutant *FLT3/ITD* and 3.6 (0.1-11.3) years for those with wild-type ($P<0.001$)”*

3. Results, 3.5 The impact of SCT...– remove “from text: After adjusting for potential confounders such as “years of diagnosis, Gender; Chemotherapy Protocol...”, our K-M survival [...]

Response: *Per the reviewer’s suggestion, we have removed “After adjusting for potential confounders such as “years of diagnosis, Gender; Chemotherapy Protocol...” in the revised manuscript.*

4. Results, 3.7 Multivariate analysis and Table 4: This is multivariable analysis, not multivariate (because it includes one outcome at a time with multiple predictor variables). Please correct.

Response: *We thank the reviewer for pointing out this error. Per the reviewer’s suggestion, we have changed “multivariate analysis” to “multivariable analysis” in Result 3.7 and Table 4 (page 10)/(page 21)*

5. Discussion: This statement over-states the robustness of results from this single study and should be modified to deemphasize the definitiveness of the allelic ratio threshold determined herein – “...this is the first time to use such a method to identify the threshold of AR, which makes the threshold credible and reliable.”

Response: *We apologize for the overstatement and robustness of this study. We fully agree with the reviewer, and therefore, we have deleted this sentence in the revised manuscript.*

5a. Same for this statement – “Interestingly, as a result, FLT3/ITD AR of 0.5 was confirmed as the prognostic threshold...”

Response: *Again, we do apologize for the overstatement. We have deleted this sentence from our revised manuscript.*

6. Figure 6 should be removed and only Table 4 included to accurately report these results.

Response: *Per the reviewer's suggestion, we have removed previous Figure 6, and include only Table 4 to accurately report these results.*

REVIEWERS' COMMENTS

Reviewer #1 (Remarks to the Author):

The authors have satisfactorily addressed all my previous concerns. The paper does need English language editing, but is otherwise satisfactory for publication. Thank you.

Reviewer #4 (Remarks to the Author):

The authors have addressed my comments and the manuscript is much improved.